

# Comparative analysis of chloroplast genomes for five *Dicliptera* species (Acanthaceae): molecular structure, phylogenetic relationships, and adaptive evolution

Sunan Huang[1,2], Xuejun Ge[1], Asunción Cano[3], Betty Gaby Millán Salazar[3] and Yunfei Deng[1]

[1] Key Laboratory of Plant Resources Conservation & Sustainable Utilization, South China Botanical Garden, Chinese Academy of Sciences, Guangzhou, Guangdong, China
[2] University of Chinese Academy of Sciences, Beijing, China
[3] Facultad de Ciencias Biológicas y Museo de Historia Natural, Universidad Nacional Mayor de San Marcos, Lima, Peru

Corresponding author
Yunfei Deng, yfdeng@scbg.ac.cn

## ABSTRACT

The genus *Dicliptera* (Justicieae, Acanthaceae) consists of approximately 150 species distributed throughout the tropical and subtropical regions of the world. Newly obtained chloroplast genomes (cp genomes) are reported for five species of *Dilciptera* (*D. acuminata*, *D. peruviana*, *D. montana*, *D. ruiziana* and *D. mucronata*) in this study. These cp genomes have circular structures of 150,689–150,811 bp and exhibit quadripartite organizations made up of a large single copy region (LSC, 82,796–82,919 bp), a small single copy region (SSC, 17,084–17,092 bp), and a pair of inverted repeat regions (IRs, 25,401–25,408 bp). Guanine-Cytosine (GC) content makes up 37.9%–38.0% of the total content. The complete cp genomes contain 114 unique genes, including 80 protein-coding genes, 30 transfer RNA (tRNA) genes, and four ribosomal RNA (rRNA) genes. Comparative analyses of nucleotide variability (Pi) reveal the five most variable regions (*trnY*-GUA-*trnE*-UUC, *trnG*-GCC, *psbZ-trnG*-GCC, *petN-psbM,* and *rps4-trnL*-UUA), which may be used as molecular markers in future taxonomic identification and phylogenetic analyses of *Dicliptera*. A total of 55-58 simple sequence repeats (SSRs) and 229 long repeats were identified in the cp genomes of the five *Dicliptera* species. Phylogenetic analysis identified a close relationship between *D. ruiziana* and *D. montana*, followed by *D. acuminata*, *D. peruviana*, and *D. mucronata*. Evolutionary analysis of orthologous protein-coding genes within the family Acanthaceae revealed only one gene, *ycf15,* to be under positive selection, which may contribute to future studies of its adaptive evolution. The completed genomes are useful for future research on species identification, phylogenetic relationships, and the adaptive evolution of the *Dicliptera* species.

## INTRODUCTION

The genus *Dicliptera* Juss. belongs to tribe Justicieae of the family Acanthaceae; it consists of approximately 150 species, which are typically found in the tropical and subtropical regions of the world (*Scotland & Vollesen, 2000*; *Mabberley, 2017*; *Hu et al., 2011*). It is readily recognized by umbellately arranged, rarely solitary, cymose inflorescence units (cymules) subtended by conspicuously paired bracts, anthers with two partially or completely superposed thecae and, in the Palaeotropics, resupinate corollas that lack a rugula (*Darbyshire, 2008*). Eleven species of *Dicliptera* are found in Peru, most of which are located in the Andes (*Brako & Zarucchi, 1993*; *León, 2006*). Species such as *Dicliptera chinensis*, *D. peruviana,* and *D. verticillata* are used in traditional herbal medicines in China and Peru (*Bussmann & Glenn, 2010*; *Horacio, Graciela & Percy, 2007*; *Telefo, Moundipa & Tchouanguep, 2002*; *Zhang, Zhu & Gao, 2010*). Species delimitation within *Dicliptera* is difficult (*Balkwill, Norris & Balkwill, 1996*) due, in part, to the remarkable uniformity of the floral morphology in the majority of the taxa. Its taxonomy is also confounded by the presence of several widespread species complexes (*Darbyshire, 2008*). The taxonomic difficulties make it important to analyze the species using molecular analysis, infrageneric classification, and the relationships within *Dicliptera. Kiel et al. (2017)* conducted the only phylogenetic analysis of the tribe Justicieae, using five chloroplast regions (*ndhF*-*trnL*, *trnT*-*trnL*-UAG, *trnS*-*trnG*, *ndhA*, *rpl16*) and one nuclear region (nr*ITS*). However, the interspecific relationships within *Dicliptera* have not been determined because of the limited number of samples available.

Five common Peruvian species were collected, representing the genus *Dicliptera* (*D. acuminata*, *D. peruviana*, *D. montana*, *D. mucronata,* and *D. ruiziana). D. acuminata*, *D. montana,* and *D. peruviana* are used in the agroindustrial industry (*Victor, Jhan & Raúl, 2017*); *D. peruviana* is a traditional herbal medicine used by the Andeans of Canta, Lima, Peru to alleviate stomach aches (*Horacio, Graciela & Percy, 2007*); *D. mucronata* is distributed mainly in Central America and is easily confused with *D. scabra* (*Victor, Jhan & Raúl, 2017*); *D. ruiziana* is found throughout Peru to elevations of about 3,000 m (*Antonio Galán et al., 2009*). All five species were collected in Southeast Peru and were distinguished from each other by the character of their leaves, bracts, bracteoles, and calyxes (Table 1). However, species delimitation is difficult without the aid of the flowers and the distribution of the five *Dicliptera* species is often overlapping. Species determinations in Central America are predominantly made by morphological comparisons as opposed to molecular comparisons. The study of the complete cp genomes of five *Dicliptera* species may encourage more effective species identification within the genus *Dicliptera*, especially in Central America.

The chloroplast genome (cp genome) is an independent genome that has been used in many evolutionary studies (*Fan et al., 2018*; *Gao, Wang & Deng, 2018*; *George et al., 2015*; *Chen et al., 2018*; *Inkyu et al., 2018*; *Kim & Lee, 2004*; *Wang et al., 2016*; *Mader et al., 2018*; *Meng et al., 2018*; *Ma et al., 2017*; *Raubeson et al., 2007*; *Wang et al., 2008*; *Wu et al., 2018*). It has a simple structure with a low molecular weight and multiple copies. Most of the cp genomes have circular structures with quadripartite organizations composed

**Table 1  Morphological differences among *Dicliptera acuminata*, *D. peruviana*, *D. montana*, *D. mucronata* and *D. ruiziana*.**

| Species | D. acuminata | D. peruviana | D. montana | D. mucronata | D. ruiziana |
|---|---|---|---|---|---|
| Plant height | Ca. 60 cm | Ca. 60 cm | Ca. 50 cm | 60–130 cm | Ca. 30 cm |
| Stem | Erect, branched, sulcate, hirsute | Erect, branched, sulcate, hirsute | Erect, branched, sulcate, pubescent | Erect, branched, sulcate, pubescent | Erect, branched, sulcate, pubescent |
| Leaf blade | Oblong-lanceolate, 3.5–7.0 × 1.5–2.5 cm, villous | Ovate, 3.5–6.0 × 2.5–4.0 cm, pubescent | Ovate, 1.0–1.5 × 0.8–1.0 cm, pilose when young, then glabrescent | Ovate, 3.0–3.5 × 1.5–2.0 cm, scarcely pilose | Ovate, 1.0–1.5 × 0.8–1.0 cm, pubescent |
| Leaf apex | Acuminate-acute | Acute | Acuminate | Acuminate-acute | Acute |
| Inflorescence | Verticillaster | Verticillaster | Spikelike thyrse | Verticillaster | Pedunculate cyme |
| Bracts | Lanceolate-linear, ciliate | Ovate, ciliate | Spatulate, gland-tipped pilose | Obovate-rhombic, pilose | Obovate, gland-tipped pilose |
| Bracteoles | Subulate, ciliate | Subulate, | Hyaline, asymmetrical, minute pilose | Linear-subulate, pilose | Subulate |
| Calyx lobes | Subulate, ciliate | Linear, hirsute | Lanceolate, minute pilose | Linear-lanceolate, margin minutely pubescent | Lanceolate, gland-tipped pilose |
| Corolla | Purple, outside pubescent | Purple, outside pubescent | Pale purple, outside pilose | Purplish red, outside pilose | Pink, outside pubescent |
| Style | scarcely pilose | glabrous | scarcely pilose | glabrous | scarcely pilose |

of one large single copy region (LSC), one small single copy region (SSC), and a pair of inverted regions (IRs). However, there are numerous exceptions to the common structure, like the IR-lacking clade (IRLC) in Fabaceae (*Fan et al., 2018*; *Gao, Wang & Deng, 2018*; *George et al., 2015*; *Chen et al., 2018*; *Inkyu et al., 2018*; *Kim & Lee, 2004*; *Wang et al., 2016*; *Mader et al., 2018*; *Meng et al., 2018*; *Schwarz et al., 2015*). The complete cp genomes of more than 2,400 plants have been published, to date, in the NCBI database (http://www.ncbi.nlm.nih.gov/genome). The majority of plant cp genomes are 110 to 170 kb in length (*Olmstead & Palmer, 1994*; *Weng et al., 2013*; *Wicke et al., 2011*). The family Acanthaceae is a large family with approximately 230 genera and 4,300 species, yet only ten species from this family have fully sequenced cp genomes (Table S1).

The complete chloroplast genome is widely used for species identification, phylogenic studies, and studies in adaptive evolution (*Wang et al., 2016*; *Ma et al., 2017*; *Fan et al., 2018*; *Gao, Wang & Deng, 2018*). Adaptive evolution is defined as the suitability for the improvement of a species during its evolutionary processes. It is always driven by evolutionary processes such as natural selection and leads to biological pressures and biodiversity at all levels of biological organization (*Yang & Swanson, 2002*; *Scottphillips et al., 2013*; *Hall & Strickberger, 2008*). The non-synonymous ($K_A$)/synonymous rate ($K_S$) ratio ($\omega = K_A/K_S$) provides a measure of selective pressure at the amino acid level. As suggested by *Makalowski & Boguski (1998)*, the $\omega$ values less than one ($K_A/K_S < 1$) indicate that the gene is under negative selection and vice versa (*Wojciech & Mark, 1998*; *Meng et al., 2018*). Recent studies have detected many positively selected chloroplast genes ($K_A/K_S > 1$), such as the *ndhC*, *ndhJ*, *psbK*, *psbN*, *rpl14*, *rpl16*, *rps4*, *rps15*, *rps18*, *rps19*, *infA*, and

*rpoB* genes in *Echinacanthus* and the *petA*, *psbD*, *psbE*, *ycf3*, *psaI*, *rps4*, *psbM*, *ndhE*, *ndhG* and *rpoC1* genes in *Allium* (*Gao, Wang & Deng, 2018*; *Xie et al., 2019*).

The cp genomes of five *Dicliptera* species were sequenced, compared, and reported for the first time in this study. The five most variable regions were identified through genome comparison analysis and nucleotide variability; these were chosen as candidate molecular markers for taxonomic identification and systematic analysis in the future. Codon usage analysis was conducted to find the codon bias in the genus *Dicliptera*. 285 simple sequence repeats (SSRs), 21 polymorphic SSRs, and 229 long repeats were detected and described. The phylogenetic relationships of the five species and other members of the family Acanthaceae were analyzed. Finally, the orthologous protein-coding genes were identified in the family Acanthaceae and the selective pressure for these genes was analyzed. This work may contribute to future adaptive evolution analysis of the Acanthaceae species.

## MATERIAL AND METHODS

### Plant materials and DNA extraction

Fresh leaf tissues were collected during several botanical surveys conducted by South China Botanical Gardens, Chinese Academy of Sciences and Facultad de Ciencias Biológicas y Museo de Historia Natural, Universidad Nacional Mayor de San Marcos in Peru. The samples were dried in silica gel immediately after collection. Voucher specimens were deposited at the Museo de Historia Natural, Universidad Nacional Mayor de San Marcos (USM) and the herbarium of South China Botanical Garden, Chinese Academy of Sciences (IBSC) (Table 2). The specimens were visually identified by Deng Yunfei and the total genomic DNA was extracted using a modified CTAB method (*Doyle & Doyle, 1987*) that included 4% CTAB with 2% polyvinyl polypyrrolidone (PVP) (*Yang, Li & Li, 2014*).

### Genome sequencing, assembly, and annotation

Short-insert (300–500 bp) libraries were constructed using the Nextera XT DNA Library Prep Kit (Illumina) following the manufacturer's instructions. Illumina × Ten instruments at BGI-Wuhan were used to perform paired-end (PE) sequencing for each sample. GetOrganelle v. 1.6.2 (*Jin et al., 2018*) was used to assemble the sequenced PE reads. *Andrographis paniculata* (GenBank accession no. NC_022451) served as a reference and the sequenced reads were filtered using Bowtie2 v. 2.3.5.1 (*Langmead & Salzberg, 2012*); SPAdes v. 3.13.1 (*Bankevich et al., 2012*) was used to assemble the filtered plastid reads and the final "fastg" files were reduced using the "slim_fastg.py" script in GetOrganelle to retain the pure plastid contigs; the filtered De Brujin graph files were transferred to Bandage v. 0.8.1 (*Wick et al., 2015*) for visualization and to obtain the paths of the final "fasta" files of the cp genomes; finally, the genome structures of all five species were compared to the reference genome using Mauve v. 1.1.1 software (*Darling, Mau & Perna, 2010*) to determine the accuracy of the final genome. The assembled cp genome was annotated using PGA v. 2019 (*Qu et al., 2019*) using the annotated *A. paniculata* as a reference. The boundaries of the annotated genes were manually modified and coupled with CDSs in Geneious v. 2019.0.3 (*Kearse et al., 2012*). All transfer RNA (tRNA) genes were determined using tRNAscan-SE v. 2.0 (*Schattner, Brooks & Lowe, 2005*). The annotated cp genome files were submitted

**Table 2** Species information and genome features of the chloroplast genomes of five *Dicliptera* species.

| Species | *D. acuminata* | *D. peruviana* | *D. montana* | *D. ruiziana* | *D. mucronata* |
|---|---|---|---|---|---|
| Location | 9.05°S, 77.81°W | 11.41°S, 77.23°W | 11.79°S, 77.05°W | 15.87°S, 74.15°W | 12.21°S, 76.82°W |
| Geographic region | Caraz, Paron, Peru | Lomas de lguanil, Huaral province, Peru | Lomas de Carabayllo, Lima province, Peru | Lomas de Chá-parra, Caravelí province, Peru | Santuario del Amancay, Lima Province, Peru |
| Voucher specimens No. | P10-091 | P170099 | P170177 | P170209 | P170492 |
| Assemblied reads | 3745800 | 2103300 | 2272793 | 2108293 | 1927500 |
| Mean coverage | 3727.5 | 2092.0 | 2262.4 | 2097.8 | 1918.3 |
| Size (bp) | 150738 | 150811 | 150689 | 150750 | 150720 |
| LSC length (bp) | 82844 | 82919 | 82796 | 82843 | 82834 |
| SSC length (bp) | 17092 | 17090 | 17091 | 17091 | 17084 |
| IR length (bp) | 25401 | 25401 | 25401 | 25408 | 25401 |
| CDSs total length | 78714 | 78714 | 78714 | 78747 | 78717 |
| Number of total genes | 114 | 114 | 114 | 114 | 114 |
| Number of Protein-coding genes | 80 | 80 | 80 | 80 | 80 |
| Number of tRNA genes | 30 | 30 | 30 | 30 | 30 |
| Number of rRNA genes | 4 | 4 | 4 | 4 | 4 |
| Overall GC content (%) | 38.0% | 38.0% | 38.0% | 38.0% | 38.0% |
| GC content in LSC (%) | 36.0% | 36.0% | 36.0% | 36.0% | 36.0% |
| GC content in SSC (%) | 31.9% | 31.7% | 31.9% | 31.9% | 31.6% |
| GC content in IR (%) | 43.3% | 43.4% | 43.3% | 43.3% | 43.4% |
| Genbank accession number | MK830556 | MK833945 | MK833946 | MK833947 | MK848596 |

to OGDRAW v. 1.3.1 (https://chlorobox.mpimp-golm.mpg.de/OGDraw.html) (*Greiner, Lehwark & Bock, 2019*) to create a circular cp genome map for each species. The five cp genomes (*D. acuminata*, MK830556; *D. pruviana*, MK833945; *D. montana*, MK833946; *D. ruiziana*, MK833947; *D. mucronata*, MK848596) were submitted to Genbank.

## Genome comparison and structural analysis

Five cp genomes were compared using mVISTA v. 2.0 (*Frazer et al., 2004*) with Shuffle-LAGAN mode and the annotation of *D. acuminata* as a reference (*Brudno et al., 2003*). The conserved regions were visualized on an mVISTA plot. DnaSP v. 5.1 (*Librado & Rozas, 2009*) was used to calculate the nucleotide variance (Pi) within the five *Dicliptera* species. The SC and IR boundaries were compared with *A. paniculata* as a reference. The Relative Synonymous Codon Usage (RSCU) of all protein-coding genes was analyzed for each species using CondoW v. 1.4.2 (*Sharp & Li, 1987*). MISA v. 1.0 (*Thiel et al., 2003*) was used to identify the simple sequence repeats (SSRs). The locations and lengths of long repeats (including forward, palindrome, complement, and reverse repeats) were analyzed using REPuter v. 2.74 (*Kurtz et al., 2001*) with the minimum repeat size set to 20 bp. Tandem repeats were identified using Tandem Repeats Finder v. 4.09 (*Benson, 1999*).

## Phylogenetic analyses

Phylogenetic analysis was conducted for all of the sequenced cp genomes (each cp genome included only one IR), including the five species reported in this study and ten previously reported species of Acanthaceae (Table S1). *Sesamum indicum* (NC_016433) (Pedaliaceae) and *Mentha spicata* (NC_037247) (Lamiaceae) were used as outgroups. The complete cp genomes were aligned using MAFFT v. 1.3.7 (*Katoh & Standley, 2013*) and were adjusted manually as needed. The substitution models with the best fit were chosen by MrModeltest v. 2.3 (*Nylander, 2004*) based on the Akaike Information Criterion (AIC). RAxML v. 8.0.0 (*Stamatakis, 2014*) was used to reconstruct the phylogenetic relationship with the maximum likelihood (ML) method. Maximum parsimony (MP) analysis was run in Paup v. 4.0a (*Swofford, 2003*). Bootstrap values exceeding 50% were shown next to the corresponding branches. Bayesian inference (BI) analysis was conducted using MrBayes 3.2.7 (*Ronquist & Huelsenbeck, 2003*) with posterior probabilities (PP) obtained for each branch.

## Selective pressure analysis

OrthoMCL v. 2.0 (*Li, Stoeckert & Roos, 2003*) was used to find the orthologous genes for the family Acanthaceae. The sequences for each orthologous gene were aligned separately using MAFFT v. 1.3.7 (*Katoh & Standley, 2013*). The nonsynonymous ($K_A$) and synonymous ($K_S$) substitution rates were calculated using PAML v. 4.9 with the codeml program to analyze the selective pressures of every orthologous gene sequence (*Yang, 2007*). The $\omega$ value ($\omega = K_A/K_S$) was estimated using the method reported by *Yang & Nielsen (2000)*. The genes under positive selection were confirmed by computing the likelihood ratio tests (LRTs).

## RESULTS

### Chloroplast genome features

The average assemblies of the five cp genomes varied from 1918.3 to 3727.5 bp. The cp genome sequences were 150,738 bp (*Dicliptera acuminata*), 150,811 bp (*D. peruviana*), 150,689 bp (*D. montana*), 150,750 bp (*D. ruiziana*), and 150,720 bp (*D. mucronata*) in length (Table 2). Each of the sequences encoded 80 protein-coding genes, 30 transfer RNA (tRNA) genes, and 4 ribosomal RNA (rRNA) genes (Table 2 and Table S2; Fig. 1), of which three protein-coding genes had two introns and nine had one intron. The *Rps12* gene was trans-spliced because of the locations of the first exon at the LSC and the other two exons at the IRs. Six protein-coding genes (*rpl2, rpl23, ycf2, ycf15, ndhB,* and *rps7*), seven tRNA (*trnI*-CAU, *trnI*-GAU, *trnL*-CAA, *trnN*-GUU, *trnR*-ACG, *trnV*-GAC, and *trnA*-UGC), and all four rRNA (*rrn4.5, rrn5, rrn16* and *rrn23*) had two copies because of their location at the IR regions. The *rps12* gene was identified as the pseudogene in *D. mucronata* by the existence of internal stop codons.

The five cp genomes displayed a typical quadripartite structure, including a large single copy (LSC) region (range from 82,796–82,919 bp), a small single copy (SSC) region (range from 17,084–17,092 bp), and a pair of inverted repeat (IR) regions (25,401–25,408 bp; Table 2). The Guanine-Cytosine (GC) content of the cp genomes of the five *Dicliptera*
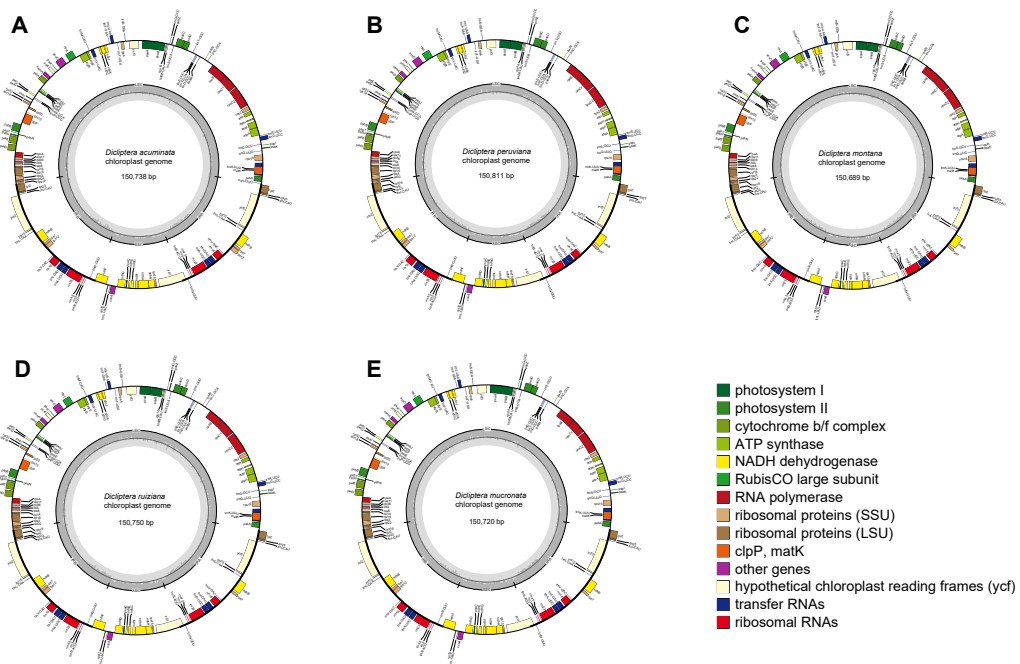

**Figure 1** **Gene maps of chloroplast genomes.** (A) *Dicliptera acuminata*; (B) *D. peruviana*; (C) *D. montana*; (D) *D. ruiziana*; (E) *D. mucronata*. Genes shown outside of the circle are transcribed clockwise, whereas genes inside of the circle are transcribed counterclockwise. The colored bars indicate known protein-coding genes, tRNA and rRNA. The dark gray area in the inner circle indicates GC content, while the light gray area indicates AT content. LSC, large single copy; SSC, small single copy; IR, inverted repeats.

was approximately 38.0%. The GC content in the IR regions (43.3–43.4%) was noticeably above that of the LSC (36.0%) and SSC (31.6–31.9%) regions in each cp genome.

## IR contraction and expansion

The IR/LSC and IR/SSC borders of the cp genomes of the five *Dicliptera* species and the *A. paniculata* were compared to identify the expansion or contraction of the *IR* (Fig. 2). The genes, *rps19*, *rpl2*, *ndhF*, *ycf1*, and *psbA* were present at the juncture of the LSC/IRa, IRa/SSC, SSC/IRb, and IRb/LSC borders. The five *Dicliptera* species have identical IR/SC borders with the exception of the *ycf1* gene at the SSC/IRb border in *D. montana,* which varies from those in *A. paniculata*. There are 167 bp from the border of *rpl2* to the juncture of LSC/IRa of *Dicliptera*, while this distance is just 57 bp in *A. paniculata*. The *rps19* gene is on the LSC/IRa border in all five *Dicliptera* species, indicating that this border has moved toward the LSC region when compared to *A. paniculata*. The *ndhF* gene in *Dicliptera* is situated at the junction of the IRa/SSC region and has 117 bp sequences located at IRa, however, the comparable region in *A. paniculata* is 40 bp long, indicating that the IRa/SSC boundary has moved toward the SSC region. The *ycf1* gene is located at the IRb/SSC junction and the border has moved toward the SSC region because there are 4,543 bp and 4,560 bp sequences situated at SSC in *Dicliptera* and *A. paniculata*, respectively. The *ycf1* gene duplications located in IRa are 811–812 bp in *Dicliptera* and 982 bp in *A. paniculata*,

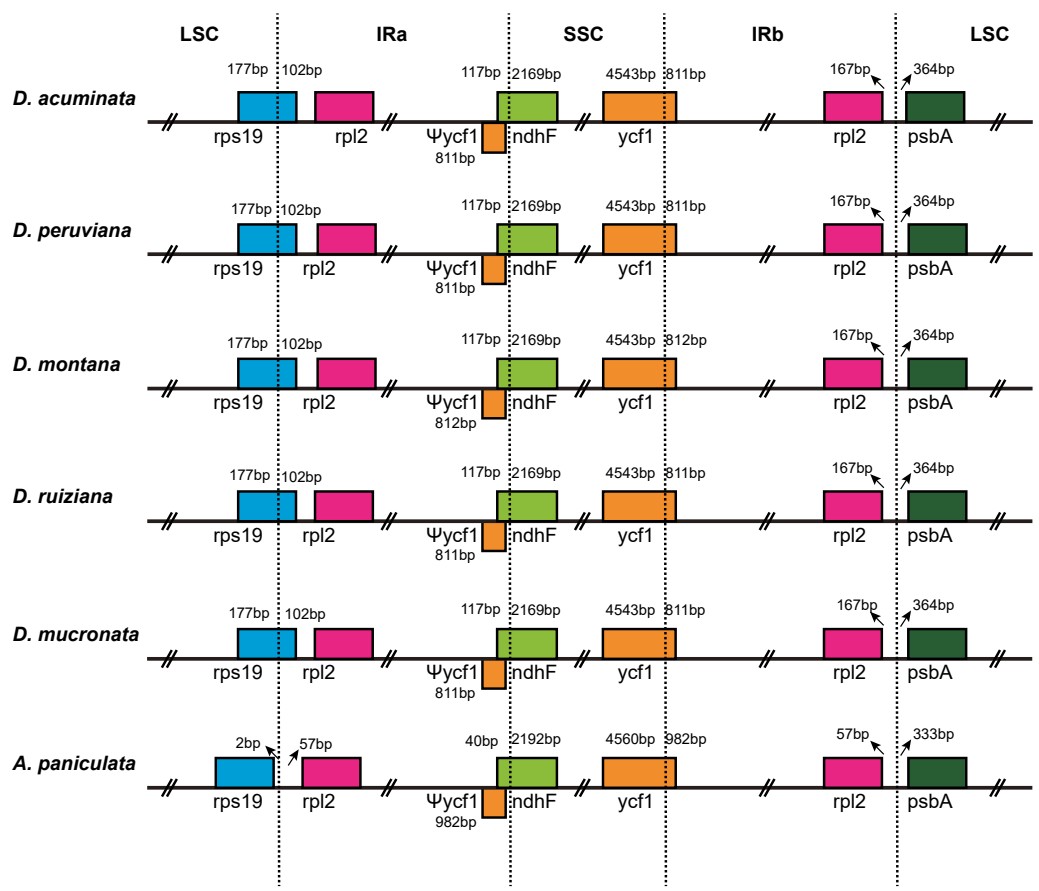

**Figure 2** **Comparison of the border regions of the LSC, SSC and IR among six Acanthaceae chloroplast genomes.** The IRb/SSC junction extended into the *ycf1* genes creating various lengths of *ycf1* pseudogenes (Ψ*ycf1*) among the six cp genomes. The number above, below or adjacent to genes shows the distance between the ends of genes and the boundary sites. The figure features are not to scale.

indicating a slight expansion of the IR regions. Likewise, the space in *Dicliptera* from *psbA* to the IRb/LSC boundary (364 bp) is enlarged compared to that of *A. paniculata* (333 bp). These findings reveal that the IR regions in the cp genome of *Dicliptera* have expanded compared to those of *A. paniculata*.

## Comparative chloroplast genome analysis

The annotated *D. acuminata* cp genome was used as a reference in mVISTA for the alignment of the cp genome among the five *Dicliptera* species (Fig. 3). The size and gene order of the chloroplast genomes of the *Dicliptera* species are conserved, but some divergent regions were identified, including the *trnH*-GUG, *rpl16*, p*etN-psbM, trnS-trnG, trnT-trnF, ndhC-trnV, petA-psbJ,* and *rps12-clpP* genes. The nucleotide variability (Pi) was calculated for the coding and non-coding regions, respectively, in order to further confirm the sequence variations (Table S3, Fig. 4). However, the Pi values are rather low among the five species (0 to 0.02230) and a total of 5 hotspot regions were identified with Pi >0.005 (*trnY*-GUA-*trnE*-UUC, *trnG*-GCC, *psbZ-trnG*-GCC, *petN-psbM,* and *rps4-trnL*-UUA).
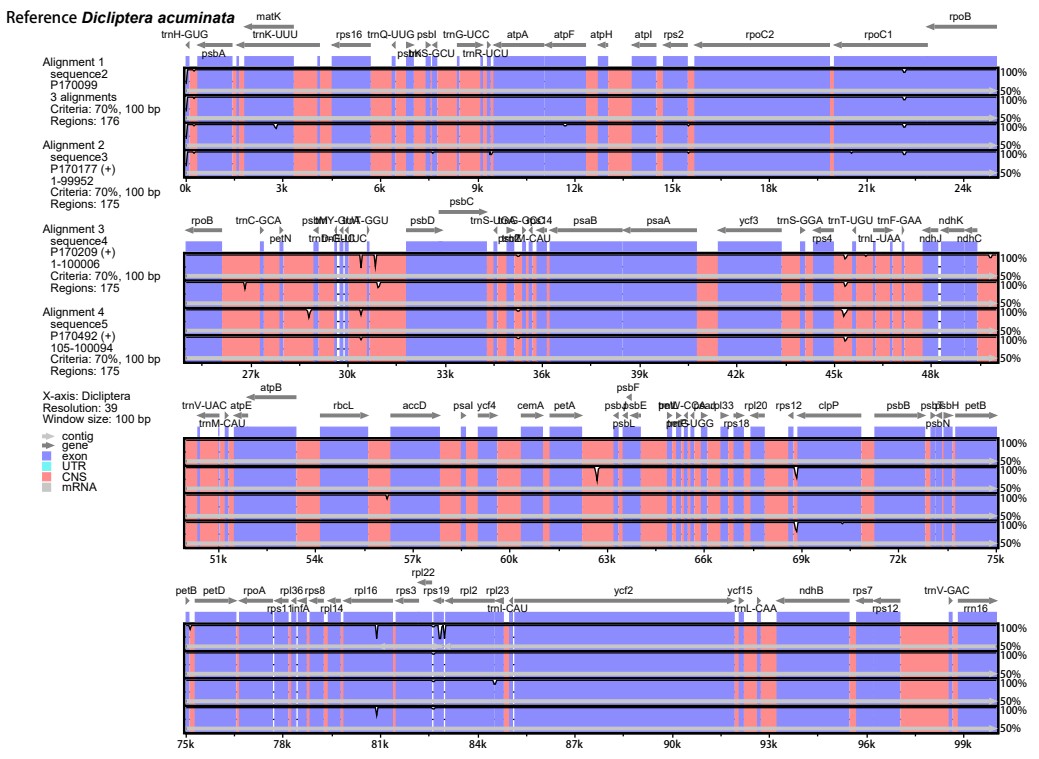

**Figure 3  Comparison of the five *Dicliptera* chloroplast genomes using mVISTA.** CNS indicates conserved noncoding sequences. The *Y*-scale represents the percent identity between 50% and 100%.

## Codon usage

A total of 78714-78747 bp protein-coding genes were identified in the five *Dicliptera* cp genomes, accounting for 52.19%-52.24% of the entire genome sequence. These genes are encoded in 26238-26249 codons. Leucine (Leu, encoded by UUA, UUG, CUU, CUC, CUA and CUG) was the most frequent amino acid encoded by these codons, comprising 2,825–2,828 (10.8%) of the total number of codons; cysteine (Cys, encoded by UGU and UGC) was the least frequently encoded amino acid, with 303–305 codons (1.2%) (Fig. 5). For all codons, from the first to third position, the AU contents are 55.8%–60.2%, 60.6%-64.1%, and 64.9%-65.8%, respectively. The majority of the preferred codons (RSCU >1) ended with A or U, with the exception of UUG (RSCU = 1.25). This phenomenon is congruent with the results from other plant studies (Table S4) (*Lu et al., 2018*; *Yang et al., 2014*; *Yi & Kim, 2012*).

## Simple Sequence Repeats (SSRs) and long repeat analysis

A total of 285 SSRs were identified in this study. The numbers of mono-, di-, tri-, and tetranucleotides were 157, 37, 55, and 36, respectively (Tables S5 and S7). Mononucleotide repeats were the most common repeats, accounting for 55.1% of the total repeats, while dinucleotides repeats accounted for 13.0%, and other SSRs occurred less frequently (Fig. 6A). The SSR varied in number and type depending on the species; *D. acuminata* and *D. montana* (58) had the most SSRs and *D. mucronata* (55) had the least (Fig. 6B, Table 3).
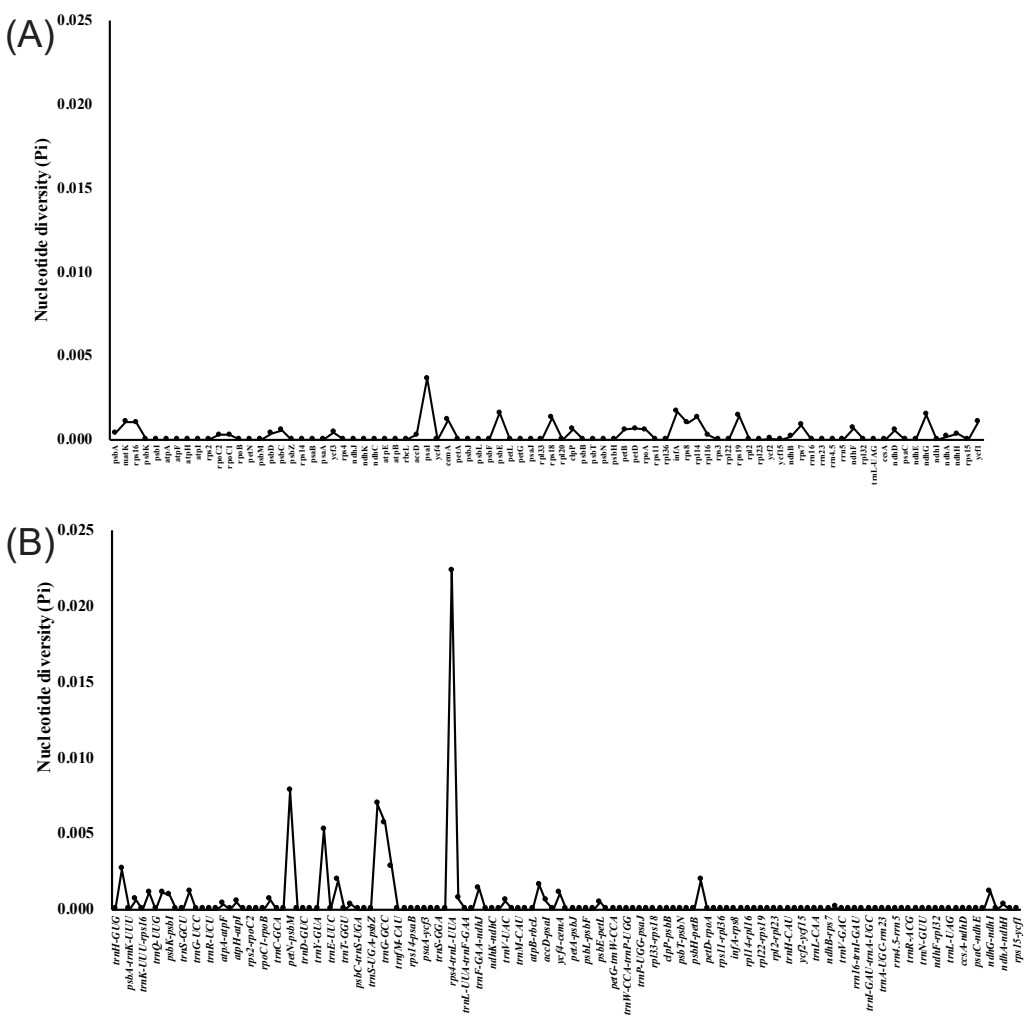

**Figure 4 Comparative analysis of the nucleotide diversity (Pi) value among five *Dicliptera* chloroplast genomes.** (A) Coding regions. (B) Non-coding regions.

Five categories of long repeats (tandem, complement, forward, palindromic and reverse repeats) were detected and analyzed in the five *Dicliptera* cp genomes (Tables S6 and S7, Fig. 6C). 229 long repeats were identified and were composed of 128 tandem repeats, 8 complement repeats, 53 forward repeats, 38 palindromic repeats, and 2 reverse repeats (Fig. 6C). The number of repeats was highest in *D. peruviana* (56) and lowest in *D. montana* (41) (Fig. 6D).

## Phylogenetic analyses

GTR and SYM+G were the best fit models used for the ML and BI trees to display the completed cp genomes. The data matrix for all of the MP, ML, and BI analyses revealed trees with highly congruent topologies. The phylogenetic relationships within the 17 cp genomes sequences analyzed were well-resolved (Fig. 7). Our phylogenetic analyses strongly support the monophyly of the *Dicliptera* species [BP$_{(MP)}$ = 100%, BP$_{(ML)}$ = 100%, PP = 1.0],

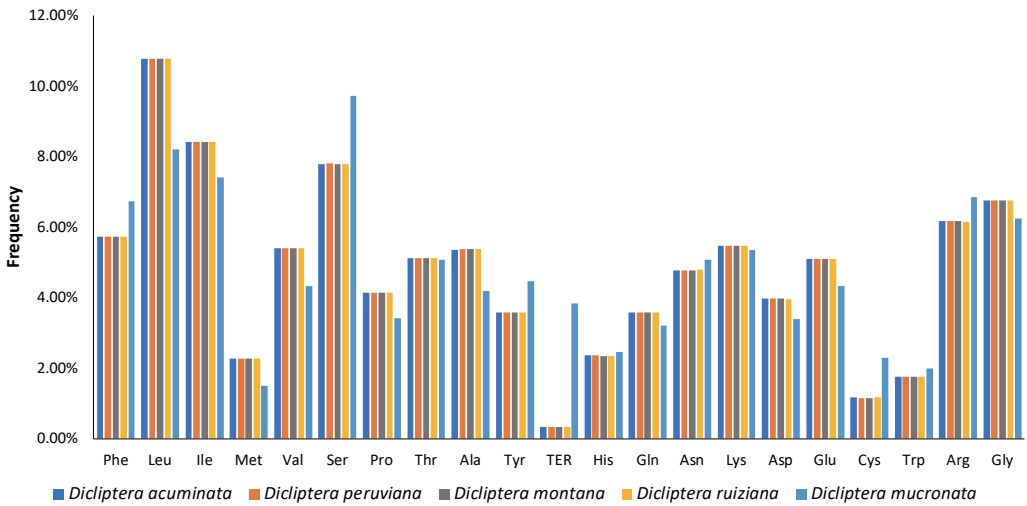

**Figure 5   Amino acid frequencies in five *Dicliptera* species protein-coding sequences.** As shown in the column diagram, Leucine was the most frequent amino acid (10.8%), Cysteine was the least (1.2%).

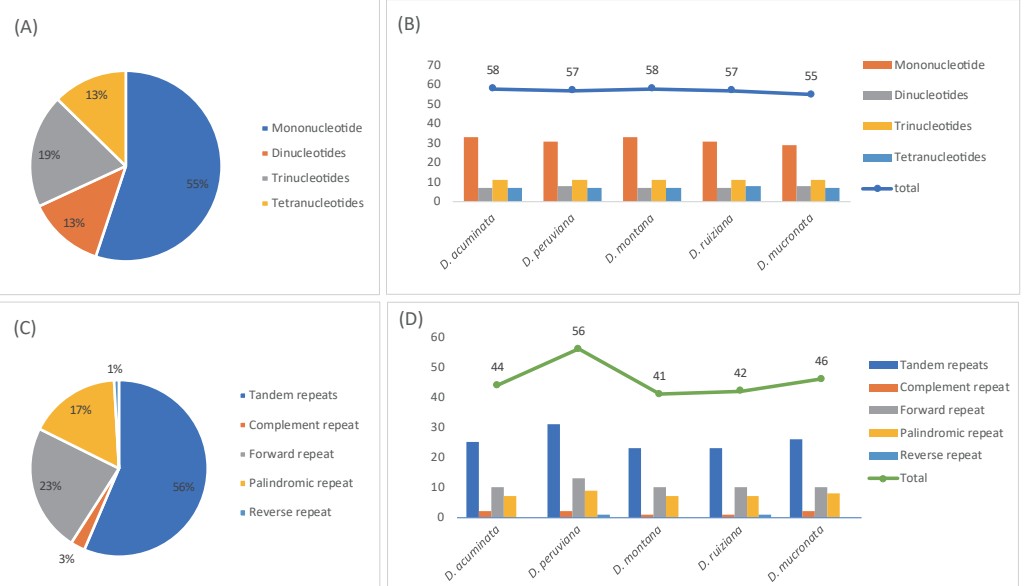

**Figure 6   The type and presence of simple sequence repeats (SSRs) and long repeated sequences in the chloroplast genomes of five *Dicliptera* species.** (A) Percentage of SSR types; (B) number of SSRs and their types; (C) percentage of five repeat types; (D) number of five repeats types.

in which *D. ruiziana* has the closest relationship with *D. montana* [$BP_{(ML)} = 52\%$, $PP = 0.96$], followed by *D. acuminata* [$BP_{(ML)}=69\%$, $PP = 0.97$], *D. peruviana* [$BP_{(MP)} = 99.8\%$, $BP_{(ML)} = 100\%$, $PP = 1.0$], and *D. mucronata* ($BP_{[MP]} = 100\%$, $BP_{[ML]} = 100\%$, $PP = 1.0$).
**Table 3** **The polymorphic SSRs among five *Dicliptera* species.** LSC, large single copy; SSC, small single copy.

| Type | *D. acuminata/D. peruviana/ D. montana/D. ruiziana/D. mucronata* | Location | Regions |
|------|------|------|------|
| A | 11/11/10/11/13 | *psbI-trnS-GCU* | LSC |
| T | 0/0/0/10/0 | *trnS-GCU-trnG-UCC* | LSC |
| T | 13/13/14/13/13 | *trnR-UCU-atpA* | LSC |
| A | 12/11/11/10/11 | *aptF* | LSC |
| A | 10/10/10/0/0 | *atpI-rps2* | LSC |
| TA | 0/0/0/0/8 | *trnE-UUC-trnT-GGU* | LSC |
| AT | 0/9/0/0/0 | *trnE-UUC-trnT-GGU* | LSC |
| T | 13/10/12/16/14 | *psbZ-trnG-GCC* | LSC |
| G | 11/11/12/9/8 | *psbZ-trnG-GCC* | LSC |
| T | 10/10/10/10/11 | *rps4-trnT-UGU* | LSC |
| ATAA | 3/3/3/6/3 | *rps4-trnT-UGU* | LSC |
| TA | 6/7/7/7/7 | *rps4-trnT-UGU* | LSC |
| T | 12/11/13/13/11 | *ndhC-trnV-UAC* | LSC |
| T | 12/11/11/12/11 | *psaI-ycf4* | LSC |
| T | 11/10/11/11/10 | *petG-trnW-CCA* | LSC |
| T | 11/10/10/10/0 | *clpP* intron | LSC |
| TA | 7/6/6/6/6 | *rpl22-rps19* | LSC |
| T | 10/10/10/10/11 | *ndhF-rpl32* | SSC |
| A | 10/11/10/10/10 | *ndhD-psaC* | SSC |
| G | 11/0/11/11/10 | *ndhG-ndhI* | SSC |
| A | 11/10/10/10/0 | *ndhA* intron | SSC |

## Selective pressure events

There were 68 orthologous protein-coding genes found in this study. The $\omega$ values of most genes were low ($\omega < 1$), approaching zero, except for the *ycf15* gene found in the SSC region, which had a ratio of 1.4453. The $\omega$ value of the *matK* gene was 0.9418, indicating a relaxed selection (Table S8, Fig. 8).

## DISCUSSION

### Sequence variation among five *Dicliptera* species

The results of our study showed that the cp genomes of five Peruvian *Dicliptera* species were similar in structure, content, and order (Table 2, Fig. 1). The cp genomes ranged in size from 150,689 bp to 150,811 bp in *D. montana* and *D. peruviana*, respectively. These structures are longer than the cp genome of *A. paniculata* (15,249 bp) (*Ding et al., 2016*). The genome size of all *Dicilptera* is relevant to LSC variation (Table 2) and this phenomenon has also been identified in other species (*Zhao et al., 2018*; *Li, Zhao & Liu, 2018*; *Meng et al., 2018*). mVISTA revealed a low divergence between the genomes of the five *Dicliptera* species, suggesting that the cp genomes were conserved. The IR regions were more highly conserved than the SC regions and the coding regions were less variable than the non-coding regions, which is also found in other angiosperms (*Gao, Wang &*

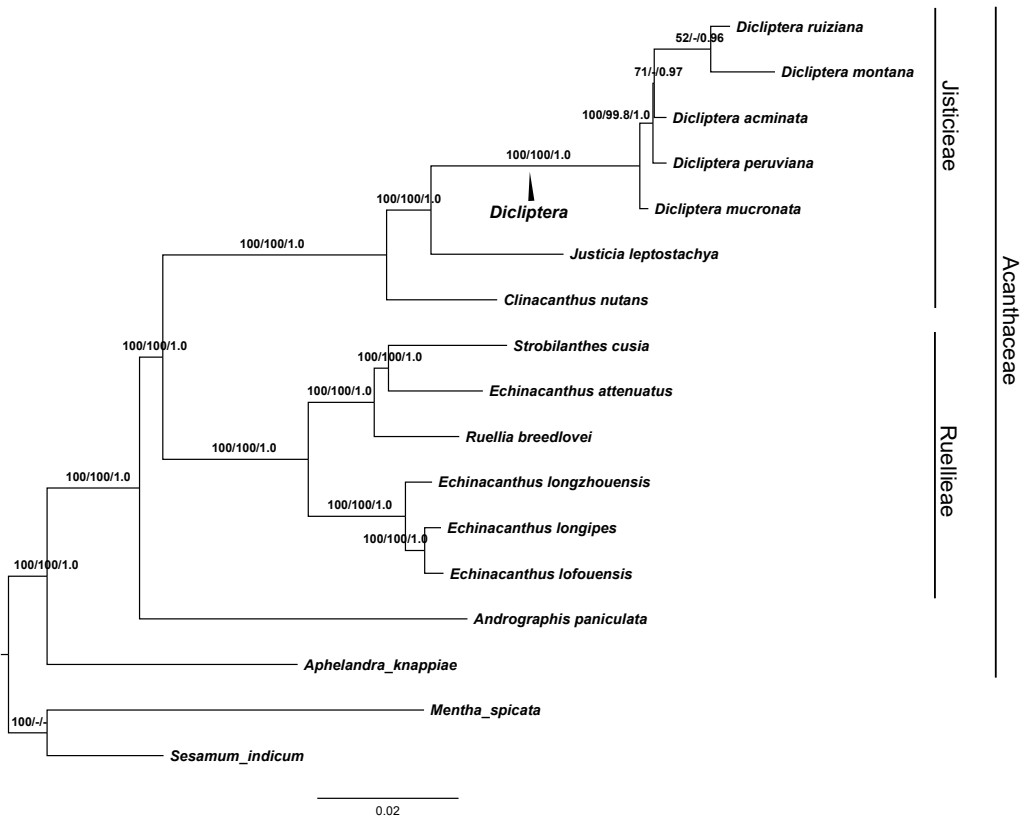

**Figure 7** **The maximum Likelihood (ML) tree of Acanthaceae.** Numbers associated with branches are ML bootstrap values, MP bootstrap values and Bayesian posterior probabilities, respecively. Hyphens indicate the bootstrap support or posterior probability lower than 50% or 0.5. *Mentha spicata* (NC_037247) and *Sesamum indicum* (NC_016433) were used as outgroups.

*Deng, 2018*; *Meng et al., 2018*; *Li, Zhao & Liu, 2018*; *Yan et al., 2019*). *Khakhlova & Bock (2006)* suggested that gene divergence with less variability in the IR and coding regions may be a result of copy corrections during gene conversion, which can correct or delete the mutation. Codons were shown to have a strong tendency toward A or U at the third codon position, which is similar to the expression of an A/U ending in other plants (*Gao et al., 2017*; *Clegg et al., 1994*; *Mader et al., 2018*; *Meng et al., 2018*). This phenomenon may explain why the Adenine-Thymine (AT) content is slightly higher than the GC content in the cp genome of *Dicliptera*.

## IR expansion analysis

IR regions are the most conserved regions in the cp genomes. Frequent expansions and contractions at the junctions of SSR and LSC with IRs illustrate the relationships among taxa and have been recognized as evolutionary signals (*Khakhlova & Bock, 2006*; *Inkyu et al., 2018*; *Lu et al., 2018*; *Raubeson et al., 2007*; *Wang et al., 2008*). In this study, only a few variations were found among the five *Dicliptera* species. When compared with the cp genome of *A. paniculata*, the IR regions of the cp genome of *Dicliptera* revealed a

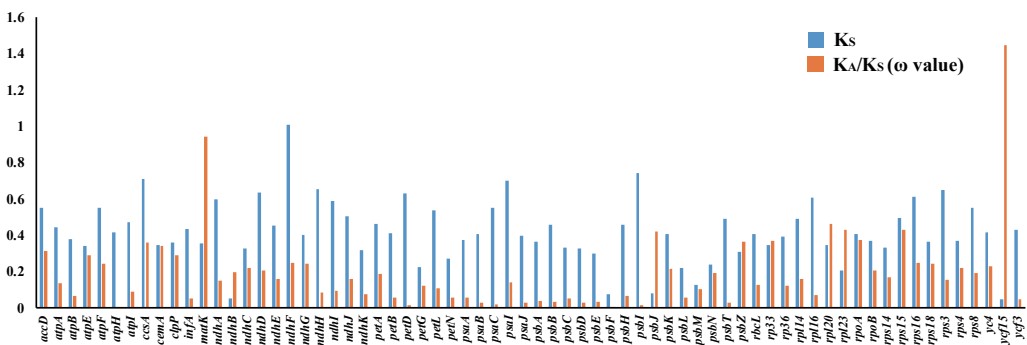

**Figure 8 Synonymous (KS) substitution rates and ω values (ω = KA/KS) among all Acanthaceae species.** As shown in the column diagram, the order of genes is alphabetical. The ω value of the *ycf15* gene (1.4453) is clearly higher than other genes.

slight expansion. The size differences among the cp genome of the five *Dicliptera* species (150,689–150,811 bp) and *Andrographis paniculata* (150,249 bp) are congruent with the results of previous studies. The contractions and expansions at the LSC/IRs and SSC/IRs junctions contribute to the size variations of the cp genomes (*Kim & Lee, 2004*; *Raubeson et al., 2007*). Gene conversion during speciation is thought to be responsible for small IR expansions or contractions. (*Wang et al., 2008*; *Goulding et al., 1996*; *Khakhlova & Bock, 2006*; *Meng et al., 2018*; *Choi, Jansen & Ruhlman, 2019*).

## Molecular markers

Simple sequence repeats (SSRs), known as microsatellites, are short stretches of DNA which consist of only one, or a few, tandemly repeated nucleotides. Polymorphic SSRs are the same units with different unit numbers located in the homologous regions; these are frequently used to identify variable species complexes (*Diethard & Manfred, 1984*; *Jerzy & Charit, 1995*; *George et al., 2015*; *Gao, Wang & Deng, 2018*). 21 SSRs were identified as polymorphic SSRs among the five *Dicliptera* species; these may be used as candidate genetic markers for further phylogenetic studies in the genus *Dicliptera* (Table 3). The presence of these repeats indicates that these regions are important hotspots for genome recombination. All polymorphic SSRs are located in LSC/SSC regions. Polymorphic SSRs are mainly distributed in non-coding regions, which are also highly variable regions in the chloroplast genomes (*Asaf et al., 2017*). The presence of long sequence repeats are indicators of mutational hotspots (*Borsch & Quandt, 2009*; *Jiang et al., 2018*).

The *ycf1* gene was previously reported for its use in DNA barcodes due to its abundance of variable sites (*Kurt et al., 2008*; *Gernandt et al., 2009*; *Dong et al., 2012*; *Drew & Sytsma, 2013*; *Dong et al., 2015*). *Shingo et al. (2013)* concluded that the *ycf1* gene is crucial for plant viability because it encodes the *Arabidopsis* protein, Tic214, which is essential for photosynthetic protein import. A substantial size difference was noted between the *ycf1* gene of the five *Dilicptera* species (5,354–5,355 bp) and *A. paniculata* (5,542 bp). The nucleotide variability of the *ycf1* gene (Pi = 0.0109) was slightly higher than that of the regions *matK* (Pi = 0.00107) and *rpl16* (Pi=0.00103). The two regions are currently used in
the DNA barcodes for the tribe Justicieae and other angiosperms (*Kiel et al., 2017*; *Särkinen & George, 2013*). Therefore, the *ycf1* gene should be a potential molecular marker for the *Dicliptera* species as well. The most divergent regions among the *Dicliptera* species, as determined by a comparison of nucleotide variability, are *rps4-trnL*-UUA (Pi = 0.02230), *petN-psbM* (Pi = 0.00783), *psbZ-trnG*-GCC (Pi=0.00697), *trnG*-GCC (Pi = 0.00571), and *trnY*-GUA-*trnE*-UUC (Pi = 0.00526). The variability in these regions was much higher than that in the coding regions and the highly variable regions identified here could be validated and used as molecular markers in future species delimitation and phylogenetic studies.

## Phylogenetic analyses

The phylogenetic trees (MP, ML and BI) demonstrated a significant relationship among Acanthaceae with high bootstrap values and posterior probabilities (Fig. 7). The genus *Aphelandra* was found to be the earliest diverging lineage; tribes Justicieae and Ruellieae are strongly supported as monophyletic groups [$BP_{(MP)}$ = 100%, $BP_{(ML)}$ = 100%, PP = 1.0] that form sister groups with each other. The results are consistent with previous studies (*Mcdade, Daniel & Kiel, 2008*; *Huang, Deng & Ge, 2019*). Phylogenetic analysis strongly supports *Dicliptera* as a monophyletic group. The clade formed by all five *Dicliptera* species is a sister to the species *Justicia leptostachya* [$BP_{(MP)}$ = 100%, $BP_{(ML)}$ = 100%, PP = 1.0], which supports the conclusion by *Kiel et al. (2017)* that the genus *Dicliptera* should be placed in the justicioid lineage. *D. mucronata* and *D. peruviana* are the first and second diverging clades among the five *Dicliptera* species; *D. acuminata*, *D. ruiziana*, and *D. montana* are species that can confidently be assigned to one clade. Trees with the same topology were retrieved from the ML and BI analyses. *D. ruiziana* was most closely related to *D. montana*, followed by *D. acuminata*. However, the relationships among *D. acuminata*, *D. ruiziana* and *D. montana* were not resolved using MP analysis. The sister relationship between *D. ruiziana* and *D. montana* is supported by their shared morphological characteristics, including a lanceolate calyx and ovate leaves of 1.0–1.5 × 0.8–1.0 cm versus the has subulate calyx and oblong-lanceolate leaves of 3.5–7.0 × 1.5–2.5 cm of *D. acuminata* (Table 1).

## Adaptive evolution analysis

Positively selected genes are known to play a key role in adapting to different environments (*Wang et al., 2016*; *Ma et al., 2017*; *Fan et al., 2018*; *Gao, Wang & Deng, 2018*; *Wu et al., 2018*) and it is important to understand the adaptive evolutionary history of Acanthaceae. Orthologous genes are a particular class of homologous genes that diverged following the speciation of their host species; they are ideal markers for analyzing evolutionary history (*Gargaud et al., 2015*). 68 protein-coding genes were found to be orthologous in the family Acanthaceae and the selective pressure of these genes was measured. The resulting measurements found that most genes in the family Acanthaceae were under negative selection ($\omega$ <1) except for *ycf15* ($\omega$ = 1.4453). According to previous studies, the *ycf15* gene is a member of the PFAM protein family accession PF10705 (*Sara et al., 2018*) and was not considered to be a protein-coding gene because of its unknown function (*Steane, 2005*; *Feng et al., 2018*). The *ycf15* gene acts as a pseudogene in some species because of

its premature stop codons (*Chen et al., 2018*; *Jiang et al., 2018*). The *ycf15* gene should be further investigated for its role in adaptive evolution and gene function.

## CONCLUSIONS

Our study sequenced and analyzed the complete cp genomes of five Peruvian *Dicliptera* species (*D. acuminata*, *D. peruviana*, *D. montana*, *D. ruiziana,* and *D. mucronata*) for the first time. The identification of the chloroplast genomes and the new molecular markers of these five species contributes to the genetic resources available for future identification and phylogenetic studies. The goal of this study was to determine the appropriate DNA barcode for the identification of the Dicliptera species, especially those found in Peru. The genes *ycf1*, *rps4-trnL*-UUA, *petN-psbM, psbZ-trnG*-GCC, *trnG*-GCC, and *trnY*-GUA-*trnE*-UUC were found to be the most suitable DNA barcode for the species *Dicliptera*. The interspecies relationships among the five species were resolved. However, further phylogenetic analysis using additional genes from the nucleus will have to be conducted in order to understand how gene introgression and hybridization affects the phylogeny of *Dicliptera* (*Birky, 1995*; *Meng et al., 2018*; *Lu et al., 2018*). A single gene, *ycf15,* was found to be positively selected among all of the protein coding genes that were identified. This gene may play an important role in the adaptive evolution of the Acanthaceae species and its function should to be further studied. Our genome data enhances the cp genome resources for the family Acanthaceae and our understanding of its species identification, phylogeny, and evolutionary history.

### Funding

This work was financially supported by the International Partnership Program of Chinese Academy of Sciences (Grant No. 151644KYSB20160005, GJHZ1620), the National Natural Science Foundation of China (Grant no. 31470302, 31670191), and ''One-Three-Five'' Strategic Planning of SCBG, CAS to Yunfei Deng and Xuejun Ge. The funders had no role in study design, data collection and analysis, decision to publish, or preparation of the manuscript.

### Grant Disclosures

The following grant information was disclosed by the authors:
International Partnership Program of Chinese Academy of Sciences: 151644KYSB20160005, GJHZ1620.
National Natural Science Foundation of China: 31470302, 31670191.

### Competing Interests

The authors declare there are no competing interests.
## Author Contributions

- Sunan Huang conceived and designed the experiments, performed the experiments, analyzed the data, prepared figures and/or tables, authored or reviewed drafts of the paper, and approved the final draft.
- Xuejun Ge conceived and designed the experiments, authored or reviewed drafts of the paper, and approved the final draft.
- Asunción Cano and Betty Gaby Millán Salazar performed the experiments, prepared figures and/or tables, apply the collecting permission, and approved the final draft.
- Yunfei Deng conceived and designed the experiments, authored or reviewed drafts of the paper, and approved the final draft.

## DNA Deposition

The following information was supplied regarding the deposition of DNA sequences:

The chloroplast sequences are available at GenBank: MK830556, MK833945, MK833946, MK833947, MK848596.

## Data Availability

Data is available at NCBI SRA: PRJNA531624, PRJNA531625, PRJNA531630, PRJNA531639, PRJNA531641.

## Supplemental Information

Supplemental information for this article can be found online at http://dx.doi.org/10.7717/peerj.8450#supplemental-information.

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
