# Peer review of "Comparative analysis of chloroplast genomes for five Dicliptera species (Acanthaceae): molecular structure, phylogenetic relationships, and adaptive evolution"

_PeerJ, doi:10.7717/peerj.8450_

## Round 0.1 · original submission · Major Revisions

The manuscript would benefit from some detailed revisions. The reviewers have identified numerous grammatical and spelling errors. Thus, it is important to correct these issues or make use of an editing service. In addition, the introduction is brief. It would be helpful to expand this section and provide more rationale for why these particular species were selected. The biggest issue identified relates to a paucity of structural and phylogenetic analyses. Providing more detailed analyses (as suggested by the reviewers) will not only improve the manuscript but will greatly improve our knowledge of IR evolution, for example.

Reviewer 1 ·

Basic reporting

The authors report whole plastome assemblies for five Dicliptera species to contribute to the phylogenetic reassessment of the genus. The introduction gives enough background, however, as the title mentions structure and adaptive evolution a short review of the relevant literature to these should be included. This could give the background to the IR junction changes and the positive selection of ycf15.

PeerJ prefers separate Results and Discussion sections, so I would suggest that you reformat the paper to fit this. This could make the paper easier to read. I found the conclusions too long and it mostly just repeats what is in the abstract. It should focus on how this study moves our understanding forward.

The article in general is well written but I have included some comments below to hopefully improve the text.

Experimental design

The study fits well in PeerJ’s scope and the experimental design is suitable to answer the main question of the study.

Validity of the findings

No comment

Additional comments

Line 17: I think it should be: Here we would like to report
Line 18: the genus is misspelled
Line 38: There should be a full stop after the bracket not a comma
Line 50: Please include date for the Kiel et al. reference here and delete the citation from the end of the sentence.
Line 51: I think regions instead of sequences would be more accurate. Also, it would be useful to include here which chloroplast and nuclear regions the Kiel et al. study used.
Line 55-56: This sentences is too vague to be informative. I guess you mean that if has been used in many evolutionary studies. In any case this needs a reference.
Line 57: circular instead of circle. It is not strictly true, not all plastomes are quadripartite, e.g. ILC clade in Fabaceae
Line 59: I am unclear about what you mean “plants in ecosystem”
Line 69: 5 should be written as five
Line 82: including instead of include
Lines 88-104: You could delete all instances of software, it is redundant. Also, the styling of version (v. or version) should be consistent. Bowtie2 and PGA are missing the version information.
Line 102: GenBank instead of Genebank
Line 103: I don’t think the link for GenBank is necessary.
Line 107: Did you mean conserved regions?
Lines 113-114: I don’t think you need (mono-), (di-) etc.
Line 119: Do you mean you included only one IR per genome?
Line 122: Used would be better than hired here
Line 127: As far as I can see all the support values are above 50% and are shown, so this sentence is somewhat misleading.
Line 132: Do you mean PCGs or all coding genes?
Line 134: program could be deleted
Line 136: the second Yang & Nielsen should be deleted
Line 140: Do you mean that the average coverage of the assemblies varied?
Line 141: delete all
Line 151: You could delete “similar processed”
Line 155: In the abstract the GC content is reported as 37.9-38%. In Table 1 all GC contents are 38%. Also, above would be better than beyond.
Line 164: What do you mean moved backwards? Backwards compared to what?
Line 179: You should delete “In recent years”
Line 187: less variable would be better than more invariable
Line 195-196: It would be useful to include the Pi for matK, rpl16, and trnS-G here for comparison.
Line 201: should be “which accounted for”
Line 202: Why not a range? Given the size and IR junction variation between the plastomes I think the percentage of PCGs would also be a range.
Lines 207-213: I think it would be useful to mention here why A/U at the third codon position matters
Line 215-216: The definition of SSRs is not entirely correct here. Check Tautz & Renz, 1984; Jurka & Pethiyagoda, 1995.
Line 216: have instead of has
Line 218: this should be mono-, di-, tri, and tetranucleotide
Line 230-231: This is somewhat obvious given the numbers above and I think could be just deleted.
Line 233: is instead of was
Line 238: were instead of was
Line 246: drawn instead of drawed. Also, include date for the Kiel et al. reference here and delete the citation from the end of the sentence.
Line 254: Makalowski and Boguski is missing the publication date and is also not listed in the references.
Lines 261-263: This should be present not past
Line 264: It would be useful to cite any references here, if there are any, that reports positive selection of ycf15. However, I don’t think you can say that ycf15 plays a crucial function as we don’t know what its function is. Also, I am not sure what you mean evolution ecosystem.
Line 298: obtained instead of apply
Line 299: collected instead of collecting
Figure 2: I would delete the last sentence of the caption and add (Ψycf1) after ycf1 pseudogenes
Figure 8: I would recommend changing the X-axis labels so that it shows all genes. Also, the peak between ndhE and ndhG seems to be above 1.0, which makes the last sentence of the caption incorrect.
Table 1: I would suggest including a geographic region in the location if this is possible. What does assembly reads mean? The number of reads used for each assembly? In which case it should be assembled reads. Or all of the sequenced reads?
Table 2: I am not sure why rps19 and ycf1 are indicated to have two copies as neither are fully in the IR to have two full copies like for example ndhB.

·

Basic reporting

The authors Huang et al reports the chloroplast genome of Dicliptera by selecting D. acuminata, D. peruviana, D. montana, D. ruiziana and D. mucronata. The article reports a novel work as this genus and its detailed chloroplast genomics have been overlooked. The authors shown that only one report is available at low level sequence analysis. Thus, suggesting that this work will further advance our knowledge to understand this genus. However, the manuscript suffer from various grammatical issues and typos. I would suggest to check it through a native speaker. The method part adapted for this work is a standard one. However, at some stage they say "five species sequenced" other places, its been shown only only D. acuminata (Fig 1). This has to be clarified. They can also make a combined circular map if they have really sequenced all the five species.

In addition, if you start with introduction, there is more need to emphasize on why only these five species selected, what were the bases of selecting only these say whether there are any taxonomical or morphological issues and sequencing could help to resolve that. A paragraph of rational is must to enlist the essentiality of this work. In addition, what is the importance of these species. I am sure if the authors come up with a photo of these species would be more attractive for the audience.

I think the authors can come up with more analysis to envision more insight from the available genome sequence data. A detailed phylogenetic analysis on the bases of shared genes in the five cp genomes will give a more detailed insight in the placement of the species. In addition, more set of phylogenetic analysis NJ, and MP would help to correct some instances.

Furthermore, I am not satisfied with the graphical presentations specially if you look at Fig 4, 5 6 and 8. Their quality and presentation style does not reflect much potential information. Similarly, if you see tables, table 2 and 3 can be moved to supplementary data. Table 1 is essential. However, some units are missing and for some species data has not been given.


A detailed expansion and contraction analysis of IR would be required to make sure how these species deviate. There are not much in depth analysis of repeats also made. They can use PHOBOS for this purpose. In addition, how Missing and ambiguous gene annotations were confirmed by comparative sequence analysis after a multiple sequence alignment and gene order comparison were performed.

Experimental design

mentioned

Validity of the findings

Mentioned

Additional comments

The authors Huang et al reports the chloroplast genome of Dicliptera by selecting D. acuminata, D. peruviana, D. montana, D. ruiziana and D. mucronata. The article reports a novel work as this genus and its detailed chloroplast genomics have been overlooked. The authors shown that only one report is available at low level sequence analysis. Thus, suggesting that this work will further advance our knowledge to understand this genus. However, the manuscript suffer from various grammatical issues and typos. I would suggest to check it through a native speaker. The method part adapted for this work is a standard one. However, at some stage they say "five species sequenced" other places, its been shown only only D. acuminata (Fig 1). This has to be clarified. They can also make a combined circular map if they have really sequenced all the five species.

In addition, if you start with introduction, there is more need to emphasize on why only these five species selected, what were the bases of selecting only these say whether there are any taxonomical or morphological issues and sequencing could help to resolve that. A paragraph of rational is must to enlist the essentiality of this work. In addition, what is the importance of these species. I am sure if the authors come up with a photo of these species would be more attractive for the audience.

I think the authors can come up with more analysis to envision more insight from the available genome sequence data. A detailed phylogenetic analysis on the bases of shared genes in the five cp genomes will give a more detailed insight in the placement of the species. In addition, more set of phylogenetic analysis NJ, and MP would help to correct some instances.

Furthermore, I am not satisfied with the graphical presentations specially if you look at Fig 4, 5 6 and 8. Their quality and presentation style does not reflect much potential information. Similarly, if you see tables, table 2 and 3 can be moved to supplementary data. Table 1 is essential. However, some units are missing and for some species data has not been given.


A detailed expansion and contraction analysis of IR would be required to make sure how these species deviate. There are not much in depth analysis of repeats also made. They can use PHOBOS for this purpose. In addition, how Missing and ambiguous gene annotations were confirmed by comparative sequence analysis after a multiple sequence alignment and gene order comparison were performed.

---

## Round 0.2 · Minor Revisions

As indicated by the reviewers, the manuscript has been improved. However, there are still some areas that require further attention. For example, it is recommended that a language editing service be used to improve the flow and clarity of the text. PeerJ offers such editing services, although you may also use an independent service if you wish. There is also concern relating to several of the figures (see reviewer comments) and ensuring that methods/results/discussion are provided in the appropriate areas.

Reviewer 1 ·

Basic reporting

The paper has improved from the previous version. There are a few corrections I would recommend before accepting the paper for publication. The adaptive evolution part of the introduction could be improved as it mostly consists of a definition.

Experimental design

no comment

Validity of the findings

no comment

Additional comments

Line 29: I would delete while
Line 41: Replace 'with' with 'from'
Line 48: This should be evolutionary studies
Line 75: Add the in front of five
Line 172-176: I would strongly recommend to change backwards and forwards to a direction in relation to plastome feature e.g towards LSC or towards ndhF. Backwards and forwards are ambiguous especially as isomeric versions of the plastomes are present in each cell.
Line 29 & 186: Please make sure the formatting of the tRNA genes are consistent through the text.
Line 194: should this be 78714-78747? 78715 is not divisible by three and 26238 * 3 is 78714.
Line 201-203: This should be in the discussion
Line 206-207: This should be in the discussion
Line 227-229: This should be in the discussion
Line 235-237: This is already in the methods, it does not need to be repeated here.
Line 237-239: This should be in the methods
Line 241-243: This should be in the discussion
Line 272: What does 'this' refer to?
Line 282-283: This needs a reference. Also, I am not sure this is relevant here, there are not any major rearrangements in these plastomes.
Line 289-290: Some requires more than one reference
Line 293: Significant should be changed to substantial unless there is a statistical test showing significance
Line 294: It should be paniculata
Line 311: Please check the in-line references
Line 321-324: Are there any morphological characters separating these species? The text is not clear enough to show this.
Line 333: I am not sure adaptive genes are the best choice here could just be deleted.
Line 335-336: This sentence needs a reference

Figure 7: I would recommend showing a tree with branch lengths instead of a cladogram, where all branches are equal length. Branch lengths contain very valuable information about sequence variation.

Figure 8: There should not be lines connecting KS and KA/KS measurements as this indicates changes that are not present, especially since the order of genes in the figure is alphabetical.

·

Basic reporting

I have gone through the suggested revision you made. The MS seem improved. However, I still feel the manuscript must be submitted to a professional language editor to improve the flow and readability. Some of the sentence structures still need refinement.

The 2nd paragraph in introduction still fails to improve the rational of the MS. The author should emphasize on what specific taxonomical issues were there that they have considered doing the cp genome of five species. In addition, the last paragraph just explain what they have done, where, it must state clearly what was aimed to investigated.

In addition, I would suggest that table 2 and 3 can be moved to supplementary data, specifically table 2 as most of the genes have been reported elsewhere. The figure captions must be more elaborative to explain the figures well, e.g. fig 8. Fig 5; There is no A, B, C ... demarkation on Fig 1 (circular map), the visibility has also been reduced. I am sure you can come up with more higher quality figure. In results, the phylogenetic analysis need more explanation. In methods, more information are needed for taxonomic identification? how and who identified them? In addition, I assume if some pictorial presentation of all five species is made in revision would really improve the visibility of the manuscript.

Experimental design

N/A

Validity of the findings

N.A

---

## Round 0.3 · accepted · Accept

The manuscript will be accepted, although I strongly advise the authors to revise a couple of their figure/table captions as they contain typos - at least that is how they appear in the review PDF.

Figure 8 caption - is the second line a new sentence or a continuation of the previous?
Table 2 caption - genome doesn't need a capital letter